# Diagnostic Workup and Evaluation of Patients with Prurigo Nodularis

**DOI:** 10.3390/medicines6040097

**Published:** 2019-09-26

**Authors:** Christina D. Kwon, Raveena Khanna, Kyle A. Williams, Madan M. Kwatra, Shawn G. Kwatra

**Affiliations:** 1Department of Dermatology, Johns Hopkins University School of Medicine, Baltimore, MD 21231, USA; rkhanna8@jhmi.edu (R.K.); Kwill184@health.fau.edu (K.A.W.); skwatra1@jhmi.edu (S.G.K.); 2Department of Anesthesiology, Duke University School of Medicine, Durham, NC 27710, USA; kwatr001@mc.duke.edu

**Keywords:** medical dermatology, prurigo nodularis, systematic review, itch, pruritus

## Abstract

Prurigo nodularis (PN) is a chronic inflammatory skin disease characterized oftentimes by symmetrically distributed, severely pruritic nodules. Currently, the pathophysiology of PN remains to be fully elucidated, but emerging evidence suggests that neuroimmune alterations play principal roles in the pathogenesis of PN. There are several associated etiologic factors thought to be associated with PN, including dermatoses, systemic, infectious, psychiatric, and neurologic conditions. We conducted a systematic literature review to evaluate the clinical presentation, diagnosis, and etiologic factors of PN. In this review, we discuss common differential diagnoses of PN and recommend an evidence-based, standardized diagnostic evaluation for those with suspected PN.

## 1. Introduction

Prurigo nodularis (PN) is a chronic inflammatory skin disease characterized oftentimes by symmetrically distributed, severely pruritic nodules. Currently, the pathophysiology of PN remains to be fully elucidated, but emerging evidence suggests that neuroimmune alterations play principal roles in the pathogenesis of PN. PN is also associated with several disease comorbidities such as chronic renal failure, liver disease, HIV, and malignancy [1,2,3,4]. However, there are currently no FDA approved therapies for PN, and patients are often times recalcitrant to off-label therapies. The goal of the present study is to perform a systematic review in the literature to provide evidence for specific testing and diagnostic evaluation of PN patients.

## 2. Methods

Using the Preferred Reporting Items for Systematic Reviews and Meta-Analyses (PRISMA) guidelines, we searched the PubMed, MEDLINE and Embase databases for “prurigo nodularis” or “prurigo.” All articles, including case reports and series describing prurigo nodularis were included. English language articles were included before April 8, 2019. We limited the search to articles involving human subjects. All of the results were checked for relevance and suitability. No manufacturers or authors mentioned in these reports were contacted.

## 3. Results

Our search yielded a total of 791 records with redundancy from 1947 to 2019 containing the aforementioned key words. After removing duplicates, 342 unique records remained. Ultimately, 35 primary sources were included. We included case series and reports. In these sources, the clinical presentation, potential pathophysiology, histological characteristics and etiologic factors of PN were discussed.

## 4. Diagnosis

### 4.1. History and Physical

PN is a clinical diagnosis that is based upon information gathered from a thorough history and physical examination. In regard to history, patients will complain of severe itch that can be continuous, paroxysmal or sporadic. In addition to pruritus, patients may also describe the sensations as a burning or stinging [5].

PN characteristically presents with excoriated, hyperkeratotic, and pruritic dome-shaped nodules, papules, or plaques that are often bilaterally distributed on the extensor surfaces of the extremities (Figure 1) [6]. In addition to affecting extremities, PN can also involve parts of the trunk that are accessible to scratching such as the upper back and abdomen. Patients with back involvement often present with the “butterfly” sign as they are unable to reach the central back and this area is spared, leaving a characteristic butterfly pattern [7]. The lesions can be variable in both quantity and quality. Cases range from patients having only a few lesions to several hundreds. The lesions also vary in size, from millimeters to centimeters, and color, ranging from flesh-colored to pink to brown or black.

### 4.2. Differential Diagnoses

There are several skin diseases that can mimic PN. We discuss several rare conditions that have been reported in the literature as being masqueraders of PN. Often, these diagnoses are only elucidated after the patient is refractory to treatment and worked-up further.

#### 4.2.1. Pemphigoid Nodularis

Pemphigoid nodularis is a rare variant of bullous pemphigoid that has features of both prurigo nodularis and bullous pemphigoid. Compared to PN, pemphigoid nodularis is often characterized by larger plaques often with large areas of central erosion, ulceration, and/or blistering [8,9]. On biopsy, there is also evidence of subepidermal clefting and direct immunofluorescence (DIF) displays linear deposition of IgG and C3 at the basement membrane zone. Furthermore, patients will have circulating antibasement membrane zone autoantibodies [10,11,12].

#### 4.2.2. Actinic Prurigo

Actinic prurigo is a rare type of photodermatosis presenting with acute eruptions of severely pruritic papules or nodules often accompanied by cheilitis and conjunctivitis. This condition is more common in young girls and they present with extreme photosensitivity to UVA and UVB [13].

#### 4.2.3. Epidermolysis Bullosa

Epidermolysis bullosa (EB) has multiple variants such as EB pruriginosa and acquista that can sometimes present with nodular prurigo-like lichenified lesions seen in PN [14]. Diagnosis is often based on immunofluorescence showing antibodies against type VII collagen in the sublamina densa [15].

#### 4.2.4. Hypertrophic Lichen Planus

Hypertrophic lichen planus (HLP) can also resemble PN clinically, presenting as hyperkeratotic plaques and nodules most commonly involving the shin and ankles. Histopathology can also be very similar with both demonstrating epidermal hyperplasia, hypergranulosis, compact hyperkeratosis, increased number of fibroblasts and capillaries. However, basal cell degeneration is confined to the tips of the rete ridges and band-like infiltration is often absent in HLP compared to PN [16].

#### 4.2.5. Neurotic Excoriations

Neurotic excoriations, also known as dermatotillomania, can also lead to excoriations that can have slightly raised areas that resemble lesions seen in PN. Neurotic excoriations, however, is a psychiatric condition characterized by excessive picking of the skin.

Lastly, some more common differential diagnoses to consider include the following: insect bites, scabies surrepticius, lupus erythematosus, multiple keratoacanthomas, atopic dermatitis, and psoriasis vulgaris [17,18].

## 5. Etiologic Factors and Associated Diseases

### 5.1. Dermatoses

Some studies have pointed to dermatological conditions as the predominant etiology of PN, up to 82% [5,19]. PN has been associated with a variety of dermatological conditions, most notably atopic dermatitis (up to 46%) [5,19,20,21,22]. Other dermatological conditions that have been associated with PN are cutaneous T-cell lymphoma, lichen planus, xerosis cutis, keratoacanthomas, and bullous pemphigoid [5,23,24]. One study found that 60% of its cohort of patients with PN (*n* = 80) had xerosis. In these patients with xerosis, there were higher rates of co-morbidities including diabetes and psychiatric causes, which is discussed below [21].

### 5.2. Systemic

Systemic and metabolic causes have been implicated in 38% to up to 50% of PN cases [25,26]. Some common systemic associations of PN are chronic renal failure, liver disease (chronic hepatitis B, primary biliary cholangitis, chronic autoimmune cholestatic hepatitis), HIV, thyroid disease, diabetes and malignances, specifically non-Hodgkin’s lymphoma [3,4,27,28,29,30,31,32,33,34,35,36]. A study (*n* = 16,925) by Larson et al. found that pruritus was most strongly associated with cancers of the liver, skin, and hematopoietic system [37]. Rarer malignancies that have been associated with PN are metastatic transitional cell carcinoma of the bladder and Hodgkin’s lymphoma [30,38,39]. Other less commonly associated systemic causes are gout, iron-deficiency anemia, and celiac disease [26,32,40,41].

### 5.3. Infectious

A number of infectious causes have been implicated in PN [42]. Some infectious or parasitic causes that have been reported are *Mycobacterium tuberculosis* and *mucogenicum*, *Ascaris lumbricoides*, *Helicobacter pylori*, *Strongyloides stercoralis*, and herpes zoster [42,43,44,45,46,47,48,49,50]. In some cases, it has been found that treatment and resolution of the infection have resolved the PN and pruritic symptoms. Although the aforementioned studies and reports have linked infectious agents with PN, there is still a lack of strong evidence for a direct causal association [42].

### 5.4. Medications

Medications have been reported as a cause of PN. There have been reports implicating mainly cancer therapy agents in cutaneous side effects [48,49]. In a study by Biswal et al., 384 out of 1000 patients undergoing chemotherapy developed cutaneous adversities. Of the 384, 0.8% (*n* = 3) developed prurigo nodularis [51]. Specifically, pembrolizumab, paclitaxel, and carboplatin have been associated with the development of PN [51,52]. With these therapy agents it is thought that the persistent activation of the immune system contributes to the pathogenesis of PN. For example, pembrolizumab is an antibody that works as an immune checkpoint inhibitor by blocking programmed death 1 protein.

### 5.5. Psychiatric

PN has been significantly associated with depression, anxiety, and dissociative experiences, all of which can cause psychogenic pruritus [1,53,54,55,56,57,58]. This psychogenic pruritus is then thought to lead to PN. However, psychogenic pruritus must be distinguished from neurotic excoriations (also known as dermatotillomania), which is characterized by excessive scratching or picking of the skin, leading to skin lesions. Of note, dermatotillomania is categorized as an impulse disorder and is frequently associated with other primary psychiatric impulse disorders [59].

### 5.6. Neurologic

Neuropathic itch is a pathological condition due to some neuronal or glial damage. It has many causes including local nerve fiber compression or degeneration. Although it is more common to involve the peripheral nervous system (PNS), neuropathic itch can also stem from damage within the central nervous system (CNS) [60]. Proximal PNS etiologies are polyneuropathies, post-herpetic neuralgia, brachioradial pruritus, notalgia paraesthetica, and other entrapment neuropathies [61]. Distal PNS etiologies include small-fiber neuropathies, sensitive skin, or post-burn itch [61].

## 6. Evaluation

### 6.1. Biopsy

Although PN is a clinical diagnosis, biopsies are often warranted for lesions that do not respond to first line therapies or those with secondary complications such as bleeding or ulceration. Weigelt et al. found that PN (*n* = 136) had characteristic features such as presence of thick orthohyperkeratotis, the hairy palm sign (folliculosebaceous units seen with a thick and compact cornified layer), irregular epidermal hyperplasia, hypergranulosis, fibrosis of the papillary dermis and increased fibroblasts and capillaries. However, these histologic characteristics often coincide with other scratch-induced conditions, such as lichen simplex. Thus, correlation of clinical and histological findings is still required to distinguish PN from other scratched-induced conditions [62].

Immunohistochemical studies have shown reduced intraepidermal nerve fiber density in lesional and nonlesional skin in patients with prurigo nodularis [63]. This observed hypoplasia has been suggested to indicate subclinical small fiber neuropathy PN [63]. However, recent studies have shown that there is no functional small fiber-neuropathy and the altered epidermal neuroanatomy is thought to be a consequence of chronic scratching [64]. As a result, we do not currently recommend routine intraepidermal nerve fiber density (IENFD) testing in patients unless underlying small fiber neuropathy is suspected. Direct immunofluorescence (DIF) is helpful for distinguishing PN from autoimmune diseases, such as bullous pemphigoid, pemphigoid nodularis, and epidermolysis bullosa, which may present similarly, or even concomitantly, with PN [23].

### 6.2. Laboratory Evaluation

As previously mentioned, PN can often have an associated systemic etiology. A thorough laboratory work-up should be pursued, especially for patients without a history of underlying dermatoses, such as PN arising secondary to atopic dermatitis. Systemic causes of chronic pruritus should be particularly evaluated for common systemic etiologies, such as chronic renal failure, liver disease, HIV, and thyroid disease. Infrequently, PN has presented in the context of gout, iron-deficiency anemia, celiac disease, non-Hodgkin’s lymphoma or infection.

Strongly suggested workup includes a complete blood cell count (CBC), complete metabolic panel (CMP), thyroid, liver and kidney function tests, HIV serology, and hepatitis B and C serologies (Figure 2). Optional workup based on clinical judgement and the patient’s clinical examination includes biopsy with a hematoxylin and eosin stain, direct immunofluorescence, chest radiograph, serum and urine protein electrophoresis, urinalysis, stool exam for ova and parasites, and iron studies.

### 6.3. Pruritic Intensity and Quality of Life

In addition to the above, evaluation should also take pruritic symptoms and quality of life into consideration given the severe pruritus that PN patients often experiences [65]. There is a wide range of different scales that have been used to assess pruritic symptoms. In terms of pruritic intensity, there are monodimensional tools that have shown high reliability and concurrent validity, such as the Visual Analogue Scale (VAS), Itch Numerical Rating Scale (Itch NRS), Worst Itch Numeric Rating Scale (WI-NRS), and Verbal Rating Scale (VRS) [65,66]. The VAS uses a 10 cm scale with endpoints marked with “0” and “10” to correlate with “no itch” and “worst imaginable itch,” respectively [65]. With the NRS, patients are asked to rate their itch intensity from 0 for “no itch” to 10 for “worst imaginable itch.” Lastly, the VRS asks patients to describe their itch symptom intensity with a list of adjectives such as “0 = no itch” and “1 = mild itch” [67].

There are also multidimensional scales to assess pruritus intensity. These include the Itch Severity Scale (ISS) and the Pruritus Grading System (PGS). These multidimensional scales encompass questions around the time of day pruritic symptoms occur, the quality, intensity and location of itch, and effects on sexual symptom and sleep [68].

In terms of assessing scratch activity and lesions, there are tools available such as the Scratch Symptom Score (SSS), Prurigo Activity Score (PAS), wrist actigraphy, and accelerometers. The SSS measures individual scratch lesions standardized against the body surface [65]. The PAS is a seven-item questionnaire designed to monitor the distribution and activity of chronic prurigo lesions [69]. Although wrist actigraphy and accelerometers may prove useful in assessing activity, they have yet to be validated and may not be easily conducted in a clinic setting [65,70].

In addition to the above scales and questionnaires that often capture a patient’s itch symptoms at one point in time, there are also several tools available to monitor the course of pruritus. These include: the Dynamic Pruritus Score, Itch-Free Days, 5-D Itch Scale, Patient Benefit Index, and ItchApp [65]. These various tools encompass the course of itch intensity and location, patient satisfaction with treatment, changing symptoms and disability over different points in time.

As previously mentioned, there are high rates of psychiatric comorbidities associated with PN. Some tools used to assess depression and anxiety include: the Beck Depression Inventory, Hospital Anxiety and Depression Scale, and Hamilton Rating Scale for Depression [71,72,73,74]. In addition to assessing for psychiatric comorbidities, chronic pruritus also negatively impairs sleep as well as quality of life (QOL). Tools to measure sleep impairment include the Athens Insomnia Scale, Stanford Sleepiness Scale, Pittsburgh Sleep Quality Index, and Epworth Sleepiness Scale [75,76,77,78].

Tools to assess quality of life commonly include the ItchyQoL, 36-item short form (SF-36) and Dermatological Life Quality Index (DLQI) [79,80,81]. The ItchyQoL questionnaire is specific to pruritic symptoms effects on the quality of life and is comprised of 22 items. Its questions cover symptoms, function, patients’ emotions and self-perception in regard to their pruritus. The SF-36 focuses on similar aspects such as limitations and emotional impairment but is also used for conditions other than chronic pruritus [81]. Lastly, the DLQI is a validated questionnaire asking about the impact of a patient’s dermatological disease and its treatment on the quality of life [80].

## 7. Discussion

Prurigo nodularis is an orphan disease with no standardized diagnostic or treatment regimen and has been understudied compared to other inflammatory skin diseases. Given the highly pruritic and chronic nature of this disease, there is a very high burden of disease, including high rates of associated anxiety and depression among PN patients [82]. These quality of life issues are compounded by the poor management of patients and lack of effective therapies. Greater attention is needed to formulate standardized diagnostic algorithms in the evaluation of PN patients. In particular, from our systematic review, we would like to highlight the need for a targeted clinical examination and laboratory evaluation to exclude other diagnoses and associated etiologic factors.

More randomized controlled studies with large patient cohorts are needed for uncovering more targeted treatments. As several agents are in the pipeline, greater recognition of PN and an evidence-based diagnostic workup is necessary for optimal patient management.

## 8. Conclusions

PN is a chronic inflammatory skin disease that often has an immense psychosocial impact on patients’ quality of life. Due to its varying clinical presentations and several mimickers, we propose a diagnostic workup for those with clinically-suspected PN. More consistent diagnoses of PN could improve future studies on PN and help discover more effective therapies for patients.

## Figures and Tables

**Figure 1 medicines-06-00097-f001:**
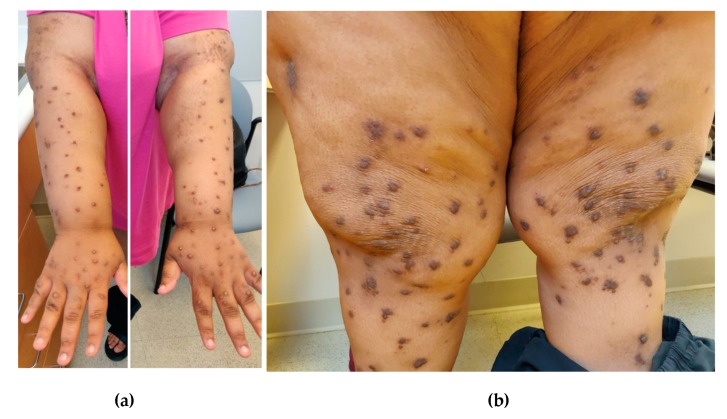
(**a**) Nodules in bilateral distribution on arms and (**b**) legs in a patient diagnosed with prurigo nodularis.

**Figure 2 medicines-06-00097-f002:**
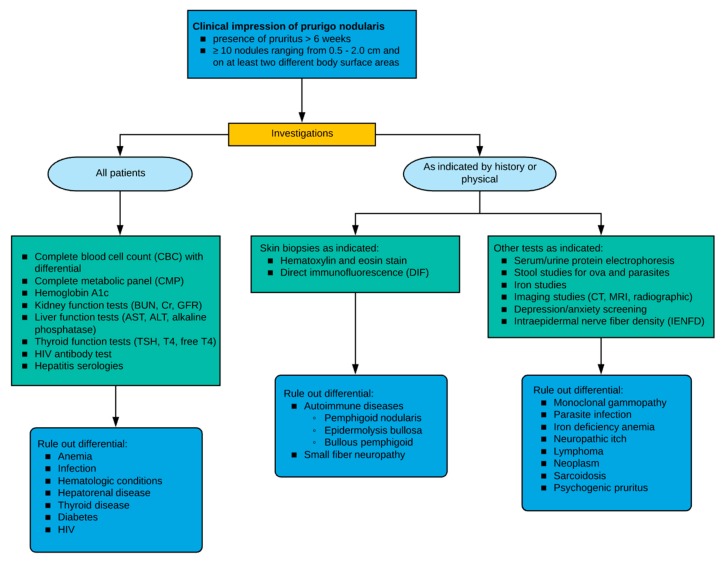
Diagnostic approach and workup for patients with prurigo nodularis.

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
