# Peer review of "Diagnostic Workup and Evaluation of Patients with Prurigo Nodularis"

_medicines, 2019, doi:10.3390/medicines6040097_

Round 1

Reviewer 1 Report

In the current paper, Christina et al elucidate evidence based recommendation for the diagnostic workup and evaluation of prurigo nodularis patients. They have used review of literature to review clinical presentations for differential diagnosis of PG.

The authors have delineated the differential diagnosis of other skin diseases which have been reported to mimic PG and have a mentioned all possible outcomes.

I would suggest to elaborate on the etiological factors and associated diseases sub heading as it seems brushed over. 

After the foloowing changes, the manuscript can be accepted.

Author Response

Thank you for this comment. We elaborated more on each section of the “Etiologic Factors and Associated Diseases” heading (lines 91-143). For the “Systemic” sub-heading, in lines 105-109 we added: “Some common systemic associations of PN are chronic renal failure, liver disease (chronic hepatitis B, primary biliary cholangitis, chronic autoimmune cholestatic hepatitis), HIV, thyroid disease, diabetes and malignances, specifically non-Hodgkin’s lymphoma [25-36]. // A study (n=16,925) by Larson et al. found that pruritus was most strongly associated with cancers of the liver, skin and hematopoietic system.” In lines 105-107, we added references to Huang et al. and Whang et al., recently published epidemiology studies on the “Real-world prevalence of prurigo nodularis and burden of associated diseases” and “Inpatient Burden of Prurigo Nodularis in the United States.”

Under the “Infectious” sub-heading, we added in lines 112-115: “In some cases, it has been found that treatment and resolution of the infection have resolved the PN and pruritic symptoms. Although the aforementioned studies and reports have linked infectious agents with PN, there is still a lack of strong evidence for a direct causal association.”

For the “Medications” sub-heading, we added the following in lines 118-128: “In a study by Biswal et al., 384 out of 1000 patients undergoing chemotherapy developed cutaneous adversities. Of the 384, 0.8% (n=3) developed prurigo nodularis [51]. Specifically, pembrolizumab, paclitaxel, and carboplatin have been associated with the development of PN [51-52]. With these therapy agents it is thought that the persistent activation of the immune system contributes to the pathogenesis of PN. For example, pembrolizumab is an antibody that works as an immune checkpoint inhibitor by blocking programmed death 1 protein.”

Lastly, in the “Psychiatric” sub-heading, we added in lines 135-136 that “Of note, dermatotillomania is categorized as an impulse disorder and is frequently associated with other primary psychiatric impulse disorders” to emphasize that these impulse psychiatric disorders often travel in groups. We hope that these revisions will provide more context for readers.

Reviewer 2 Report

This is a valuable review or systematic review on the diagnosis of prurigo nodularis.

However, there are two big concerns:

1/ The part on the evaluation is not complete because a large chapter on the evaluation of pruritus intensity and consequences on the quality of life is lacking

2/ It is not adequate to use the term "recommendations" for this paper, especially in the title, since there is no discussion in a group of experts nor support from IFSI or Prunet committees. Furthermore, the bibliography does not allow to claim that the proposals from the authors are evidence-based.

Author Response

1/ The part on the evaluation is not complete because a large chapter on the evaluation of pruritus intensity and consequences on the quality of life is lacking

Thank you for this comment. We agree that the evaluation of pruritus intensity and impact on the quality of life is a necessary component. Therefore, we have added a section entitled, “Pruritic Intensity and Quality of Life” (lines 177-214).  In this section, we outlined the various tools and scales that clinicians have used to assess pruritic symptoms, scratch activity and lesions, sleep impairment and quality of life.

2/ It is not adequate to use the term "recommendations" for this paper, especially in the title, since there is no discussion in a group of experts nor support from IFSI or Prunet committees. Furthermore, the bibliography does not allow to claim that the proposals from the authors are evidence-based.

Thank you for this suggestion. To address this, we have changed our manuscript title to: “Diagnostic workup and evaluation of patients with prurigo nodularis.” We also modified the wording in lines 232-233 to “we propose a diagnostic workup” to clarify that there was not a discussion in a group of experts for this diagnostic algoritihm. We also emphasize in lines 223-226 that “Greater attention is needed to formulate standardized diagnostic algorithms in the evaluation of PN patients.”

Round 2

Reviewer 2 Report

The manuscript has been substantially ameliorated and should be published

Please add this reference: Brenaut et al. J Eur Acad Dermatol Venereol. 2019 Jan;33(1):157-162

Author Response

Thank you for this suggested reference. We agree that the Brenaut et al. reference is an important study to include and have added it to the “Psychiatric” subsection (line 128; reference number 54) under the “Etiologic Factors and Associated Diseases” chapter.